# Conservative Management of Asymptomatic Adnexal Masses Classified as Benign by the IOTA ADNEX Model: A Prospective Multicenter Portuguese Study

**DOI:** 10.3390/diagnostics11111992

**Published:** 2021-10-27

**Authors:** Marta Espanhol Brito, André Borges, Sofia Rodrigues, Paula Ambrósio, Raquel Condeço, Abílio Lacerda, Maria José Bernardo, Patrícia Pinto, Dusan Djokovic

**Affiliations:** 1Maternidade Dr. Alfredo da Costa, Centro Hospitalar Universitário Lisboa Central (CHULC), 2890-495 Lisbon, Portugal; martafebrito89@hotmail.com (M.E.B.); sofia.ferreira.rodrigues@gmail.com (S.R.); pefsard@gmail.com (P.A.); raquelsgc@gmail.com (R.C.); abilio.lacerda@gmail.com (A.L.); mjbernardo@netcabo.pt (M.J.B.); aplpinto@gmail.com (P.P.); 2Department of Obstetrics and Gynecology, Hospital de S. Francisco Xavier, Centro Hospitalar Lisboa Ocidental (CHLO), 1449-005 Lisbon, Portugal; borges_al@hotmail.com; 3Faculdade de Ciências da Saúde, Universidade da Beira Interior, 6200-506 Covilhã, Portugal; 4Department of Obstetrics and Gynecology, NOVA Medical School—Faculdade de Ciências Médicas, NOVA University of Lisbon, 1169-056 Lisbon, Portugal; 5First Faculty of Medicine, Charles University, CZ-121 08 Prague, Czech Republic; 6Department of Obstetrics and Gynecology, Hospital CUF Descobertas, 1998-018 Lisbon, Portugal

**Keywords:** adnexal mass, benign lesions, conservative management, IOTA ADNEX model, ultrasound

## Abstract

This prospective multicentric study aiming to determine the incidence of complications (malignant transformation, torsion or rupture) during conservative management of adnexal masses was performed in two Portuguese tertiary referral hospitals. It included ≥18-year-old, non-pregnant patients with asymptomatic adnexal masses (associated IOTA ADNEX risk of malignancy < 10%) sonographically diagnosed between January 2016 and December 2020. Conservative patient management consisted of serial clinical and ultrasound assessment up to 60 months of follow-up, spontaneous resolution of the formation or surgical excision (median follow-up: 17.8; range 9–48 months). From the 573 masses monitored (328 premenopausal and 245 postmenopausal adnexal masses), no complications were observed in 99.5%. The annual lesion growth rates and increases in morphological complexity were similar in the premenopausal and postmenopausal patients. Spontaneous resolution, evidenced in 16.4% of the patients, was more common in the premenopausal group (*p* < 0.05). Surgical intervention was performed in 18.4% of the cases; one borderline and one invasive FIGO IA stage cancer were diagnosed. There was an isolated case of ovary torsion (0.17%). These data support conservative management as a safe option for sonographically benign, stable and asymptomatic adnexal masses before and after menopause and highlight the need for expedite treatment of symptomatic or increased-morphological-complexity lesions.

## 1. Introduction

Most adnexal masses are incidentally diagnosed by pelvic imaging and the vast majority is benign [1,2]. In an attempt to detect ovarian cancer in its early stages, ultrasound (US) assessment of asymptomatic women has been widely engaged [3]. However, ovarian cancer mortality does not significantly differ between screened and unscreened patients [1,4,5,6,7]. Due to the concern that detected adnexal masses can be malignant or may suffer malignant transformation [1,2,3], nearly 200,000 women undergo pelvic surgery each year in the United States alone, of which <15% are diagnosed with ovarian cancer [3,8]. This approach may lead to invasive medical procedures, iatrogenic morbidity and mortality. Associated complication rates vary between 2% and 15% [3,8]. Uncommon but well-documented major complications include infection, wound dehiscence, anesthetic complications, myocardial infarction, deep vein thrombosis, pulmonary embolism and injury to hollow viscera [4,5,6,9]. 

Patients with early diagnosed adnexal lesions may be kept under a conservative approach (i.e., clinical and imaging surveillance), depending on clinical presentation, US findings, previous medical history and patient preferences. US characterization and interpretation of adnexal masses are crucial to appropriate management. The European Federation of Societies for Ultrasound in Medicine and Biology has published minimum training requirements for gynecological ultrasound practice in Europe, identifying three levels (I, II and III) of training and expertise [10,11]. A prospective randomized controlled trial has demonstrated that level III (expert) US examinations result in a significant decrease of unnecessary major interventions when compared with level II (routine) US examinations [12]. To improve US-based discrimination between benign and malignant adnexal masses and appropriate patient triage and referral to general gynecologists vs. multidisciplinary gynecologic oncology units, standardized terminology, scanning technique and validated malignant risk prediction algorithms/models, including the logistic regression (LR) models 1 and 2, such as the Simples Rules (SR) and Assessment of Different NEoplasias in the adneXa (ADNEX) model, were developed by the International Ovarian Tumor Analysis (IOTA) group [13,14,15,16]. Subjective assessment by expert ultrasound examiners, as well as the performance of the IOTA prediction models, have already been proven to be excellent in distinguishing benign from malignant adnexal lesions [15,16,17,18,19,20]. They are increasingly accepted and used in clinical practice in many countries. Furthermore, the IOTA ADNEX model is the first prediction model that also provides the risk distribution between four malignancy categories (borderline tumors, stage I, stage II-IV primary cancers and secondary metastatic tumors in adnexal lesions) [16]. A large multicenter cohort study comparing different prediction models has demonstrated the IOTA SR and ADNEX model to be the best currently available tools [21]. Published in 2018, the Ovarian-Adnexal Reporting and Data System (O-RADS) provided a standardized lexicon with comprehensive descriptors and definitions of the US characteristic appearances of normal ovaries and different adnexal lesions [22]. The O-RADS working group has developed a patient triage system based either on the O-RADS descriptors or on the risk of malignancy determined by the IOTA ADNEX model [23]. However, the ORADS descriptors and triage system have yet to be externally validated. 

With the development of robust prediction models and their introduction into daily practice after internal and external validation, conservative management, i.e., clinical and sonographic follow-up, emerges as a potentially beneficial approach for a significant number of patients with asymptomatic adnexal formations with benign imaging features. Several retrospective studies found that most conservatively managed asymptomatic adnexal lesions remained unchanged, while many spontaneously regressed [24]. A recent, and so far the only, large prospective study conducted on long-term follow-up of asymptomatic and sonographically benign adnexal masses, the IOTA phase 5 study, reported a low risk of malignant transformation and acute complications (e.g., cyst rupture and torsion), suggesting that conservative management could be an appropriate option for asymptomatic patients with sonographically benign and stable adnexal formations [3]. 

This multicentric Portuguese project aimed to assess the US morphological evolution of asymptomatic adnexal masses diagnosed as benign according to the IOTA ADNEX model in pre and postmenopausal women. The specific objectives were to assess the rate of onset of symptoms and the incidence of complications during the clinical and sonographic follow-up period. The main hypothesis of the study was that conservative treatment is a safe option for these patients.

## 2. Materials and Methods

This prospective multicentric cohort study was conducted at the departments of obstetrics and gynecology—gynecological ultrasound units of two Portuguese tertiary referral hospitals (Maternidade Dr. Alfredo da Costa, Centro Hospital Universitário de Lisboa Central and Hospital S. Francisco Xavier, Centro Hospital Lisboa Ocidental, Lisbon, PT). It included ≥18-year-old, non-pregnant patients with asymptomatic adnexal masses (associated risk of malignancy <10%, as determined by the Assessment of Different NEoplasias in the adneXa (ADNEX) model) sonographically diagnosed during an arbitrarily chosen five-year period from January 2016 until December 2020. Of all the asymptomatic cases consecutively observed during this period, only patients with de novo diagnosed masses were included. If bilateral adnexal masses were diagnosed, each one of them was included and treated separately. Patients participating in other studies and those with evident physiological formations <3 cm were excluded (see Figure 1).

In order to assess the ultrasound (US) morphological evolution, determine the symptom appearance frequency and determinate the rate of complications during the conservative management, the follow-up visits were schedule and performed 3 and 9 months after the diagnosis, and thereafter every 12 months until spontaneous resolution, surgical excision of the adnexal masses or 60 months of follow-up. Each study participant was assessed for age, menopausal status, previous/concomitant diagnosis of breast and ovarian cancer, use of hormonal therapy and symptoms. At each follow-up visit, the onset of symptoms was assessed, physical examinations performed, transvaginal and abdominal US scan executed and blood cancer antigen 125 (CA-125) level determined. The CA-125 levels were included in the ADNEX model for determining the risk of malignancy. In the presence of new symptoms, patients were managed accordingly. 

All ultrasound examinations were performed by five gynecology specialists, professionally predominantly dedicated to gynecological ultrasound scanning and certified by the International Ovarian Tumor Analysis (IOTA) group (DD, MJB, PA, PP, RC; average experience in gynecologic ultrasonography: 11 years, range 5–30 years). The IOTA lexicon and scanning technique were used exclusively, as previously described [13]. In all cases, the scan was performed with a transvaginal and also abdominal probe, while for the purposes of the study, only measures of adnexal masses obtained by the transvaginal approach were taken into account. General Electric (GE) Voluson^TM^ E8 ultrasound devices were used to perform all exams. All ultrasound data were obtained by reading electronic ultrasound reports whose accuracy was supported by accompanying images and/or videos, stored in the hospital ultrasound databases, without detected cases of discrepancies between the images/videos and the descriptions. Persistent masses were sonographically evaluated, taking into account the increasing US complexity and lesion growth. Increased US complexity was defined as de novo detection of ≥1 solid components and/or increased number of locules. Changes related to US morphological complexity and lesion growth were assessed by comparison of the first with last performed US. Presumed histology according to the US subjective assessment was registered with the following options: simple ovarian, para-ovarian or salpingeal cyst; serous cystadenoma or cystadenofibroma; endometrioma; teratoma; functional cyst; fibroma or fibrothecoma hydrosalpinx; mucinous cystadenoma or cystadenofibroma; abscess, salpingitis, or pelvic inflammatory disease; inclusion or peritoneal cyst; rare benign tumor; or formation with sonographic characteristics that does not allow for a specific histology to be suggested.

During the study, patients underwent surgical treatment due to suspicion of malignancy (i.e., increased US morphological complexity), symptom occurrence, patient request/opportunistic reasons or fertility concerns. According to the hospital protocol, whenever malignancy was suspected, the patient was examined by thoracic, abdominal and pelvic computed tomography (CT) before the surgery, in order to plan appropriate management. Specimen histological examination was performed in all cases by two experienced pathologists of the host hospitals (both with over 15 years of specialist experience). Histological diagnosis was correlated with the patient’s preoperative clinical and US presentation.

Cases with at least 9 months of follow-up as well as all patients submitted to surgical treatment with at least one follow-up evaluation prior to surgery were selected and analyzed. For statistical analysis, the Statistical Package for the Social Sciences (SPSS) software version 24 was engaged; *p*-value < 0.05 was considered statistically significant.

## 3. Results

During the study inclusion period (60 months), 685 patients were diagnosed with 797 adnexal masses that were selected according to previous detailed inclusion criteria (Figure 1). Frequent indications for the initial US scan were uterine fibroid follow-up, subfertility, intrauterine contraceptive device control and evaluation of the adnexal formation(s) identified on a recent “routine” US assessment ordered by the primary care doctor. Of these 797 masses, 224 adnexal formations (220 patients) were loss to follow-up or had a follow-up period shorter than 9 months. The remaining 573 masses (i.e., 71.9%), including 328 adnexal formations (57.2%) observed in 260 premenopausal women and 245 masses (42.8%) diagnosed in 205 postmenopausal patients, were all analyzed. There were 54 bilateral masses (9.4%). The median follow-up time was 17.8 (range 9–48 months; standard deviation, SD: 10 months). 

The patient median age at diagnosis was 50.9 years (range 18–90 years; SD: 16 years). Other demographic features and clinical data are presented in Table 1. 

The main US features and classification of the masses, based on the US morphology and sonographer subjective assessment, are presented in Table 2. In both the pre- and postmenopausal groups, most adnexal masses were classified as unilocular (*n* = 308, 53.8%) while the most frequent presumptive histologic entities expected and reported by sonographers were endometrioma in the premenopausal group (*n* = 98, 30%) and serous cystadenoma in the postmenopausal women (*n* = 119, 48.6%). 

In most cases, no complications of any kind were observed (93.5%, Table 3). Ninety-four masses (16.4%) revealed spontaneous resolution, including 73 adnexal formations (22.3%) in premenopausal vs. 21 (8.6%) in postmenopausal women (*p* < 0.001). In this subgroup of adnexal formations, the most frequent diagnostic hypotheses (sonographer subjective assessment) were hydrosalpinx/salpingitis, simple cyst and serous cystadenoma (*n* = 73, 77.7%). The median interval between diagnosis and spontaneous resolution was 11.0 months in the premenopausal vs. 18.6 months in the postmenopausal group (*p* < 0.01, Table 4). 

Regarding persistent masses, the annual growth rate and increased complexity did not significantly differ between premenopausal and postmenopausal women (Table 5). One hundred and three (18%) adnexal masses were surgically removed (Table 3), mostly due to patient request or opportunistic reasons (*n* = 48, 46.6%). There was an isolated case of adnexal torsion (Table 3), confirmed intraoperatively (explorative laparoscopy with unilateral adnexectomy of a histologically diagnosed endometrioma). Twenty-eight masses (4.9%) underwent surgery due to increase in morphological complexity and inherent suspicion of malignancy (Table 3 and Table 6). In two cases (2/573, i.e., 3%), presented in Figure 2 and Figure 3, histological analyses of surgical specimens indicated an ovarian border-line and a mucinous ovarian cancer FIGO stage IA, respectively.

## 4. Discussion and Conclusions

The results of the present study lend support for conservative management as a safe option for sonographically benign, stable and asymptomatic adnexal masses, before and after menopause, while the onset of symptoms and increased morphological complexity of lesions should always be valorized and adequately managed. In our series, adnexal formations with the ADNEX risk of malignancy <10% remained sonographically unchanged in the vast majority of cases (*n* = 551, 96.2%). A significant proportion of these formations showed spontaneous resolution (*n* = 94, 16.4%), while complications occurred in 0.5% (*n* = 3). In accordance, in previous studies, the removal of persistent ovarian cysts was not found to decrease the ovarian cancer mortality over a prolonged observation period of 15 years [25]. 

When compared with our study, other retrospective studies with fewer cases have provided similar information [16,17,18]. Valentin and Akrawi studied the evolution of 134 conservatively managed asymptomatic postmenopausal patients diagnosed with 160 adnexal cysts with benign ultrasound features and reported surgical excision in 9%, spontaneous resolution in 29%, appearance of additional adnexal formations in 13% and stable or decreasing US complexity in 83.6% of patients, without a documented case of ovarian malignancy [24]. The higher incidence of spontaneous resolution may be explained by the higher number of included unilocular functional cysts. Furthermore, Alcázar et al. conservatively managed 120 asymptomatic premenopausal women with sonographically benign ovarian cysts < 6 cm (median follow-up: 42 months) and also observed that most lesions remained unchanged, both in size and sonographic appearance; the rate of spontaneous resolution was 8.3% and no patient developed any symptom or presented US findings suggestive of ovarian cancer [26]. 

Regarding specific US morphologic features, Castillo et al. described the evolution of simple, unilocular adnexal cysts in asymptomatic postmenopausal women during a median follow-up time of 48 months; nearly half of the adnexal lesions resolved spontaneously and most of the persisting masses remained unchanged, while the rate of malignancy was very low [27]. In the case of lesions sonographically suggestive of mature teratomas, the risk of malignancy and the risk of adnexal torsion have also been found to be very low [28,29]. Alcazar et al. studied benign-appearing purely solid ovarian lesions in postmenopausal women. Of the 99 patients included in that study, 42 women (42.4%) underwent surgery after the US diagnosis; 2 cases of ovarian primary cancer were diagnosed [19]. The remaining conservatively managed lesions (57.6%) did not change size or US morphological appearance during the mean follow-up period of 36 months. In our study, a similar behavior of solid ovarian lesions was observed in postmenopausal women.

In the preliminary report of the prospective multicenter cohort IOTA 5 study, 5 out of 1919 included patients with follow-up ≥2 years (<1%) were diagnosed with ovarian cancer, 5 (<1%) with a borderline ovarian tumor and 2 (<1%) with ovarian metastases [3]. A low risk of acute complications, such as torsion and cyst rupture, was reported, with spontaneous resolution evidenced in 20.2% and surgical intervention performed in 16.1% [3]. Thus, both the IOTA 5 and our results support the adequacy of careful monitoring instead of prompt surgical removal of every apparently non-physiologic adnexal lesion. In our study and IOTA 5 series, most surgeries were performed in premenopausal women. In this patient subpopulation, endometriomas frequently become symptomatic or require surgical intervention based on fertility concerns [3]. In line with the IOTA5 study, we also identified that the main reason for performing surgery was patient desire even in the absence of symptoms or suspicious US findings [3]. 

The importance of short time intervals between scans was evidenced in our patients diagnosed with ovarian malignancy, which were not initially adequately interpreted since the lesions did not show malignant US characteristics at their early stages of development. Importantly, ovarian cancers may also develop in apparently normal ovaries and not in (known) adnexal cysts under follow-up [24,30]. In the IOTA5 study, all but one of the diagnosed malignancies (a borderline tumor) were diagnosed and surgically removed during the first year of follow-up and nine of them were diagnosed/removed in the first 6 months [3]. The interval time between observations should be better addressed in further research projects, but cautiously—the initial frequency of clinical and US observations should be higher and then progressively reduced, in order to achieve optimal safety of the conservative approach.

To sum up, a growing body of evidence, including the data presented here, indicates that expectant management is a safe option for asymptomatic, incidentally detected and morphologically stable adnexal masses characterized as benign by the IOTA ADNEX model, both in premenopausal and postmenopausal women. The onset of related symptoms and/or increasing US complexity of the lesion should receive immediate attention, guaranteeing timely and adequate management of the suspected cases. Future research, including large-scale multicenter studies, should enable the establishment of precise monitoring protocols for different malignancy risk cut-offs, with the ultimate goal of facilitating patient counseling and contributing to the adoption of appropriate personal attitudes in patients. 

## Figures and Tables

**Figure 1 diagnostics-11-01992-f001:**
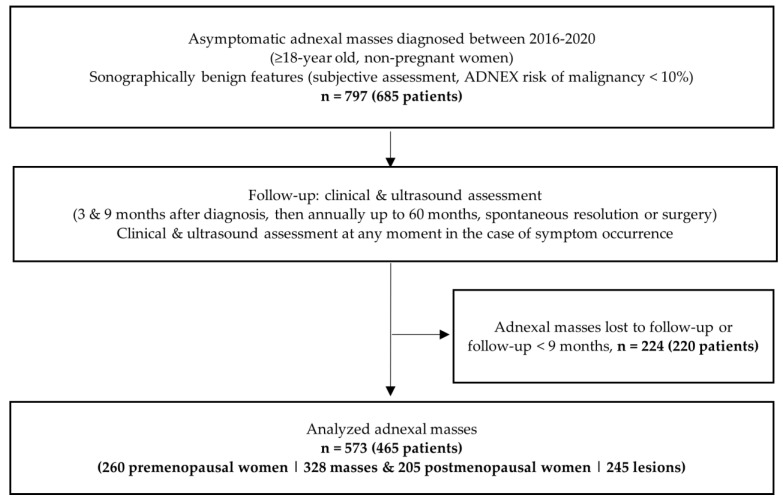
Study Design.

**Figure 2 diagnostics-11-01992-f002:**
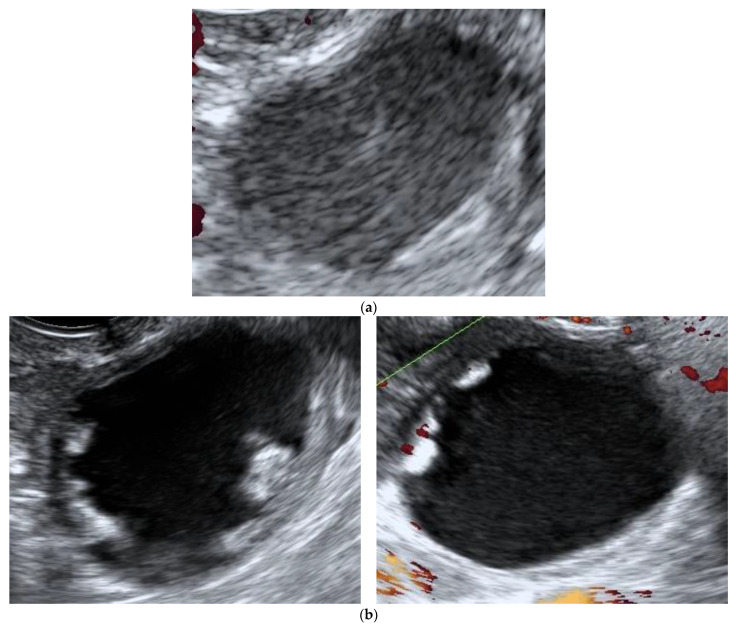
Ultrasound features of borderline ovarian tumor initially classified as a benign lesion. (**a**) Unilocular ovarian formation with “ground glass” content and color score 1 with 35 × 30 × 20 mm, observed in an asymptomatic post-menopausal woman (age: 52 years), classified as a benign by the IOTA ADNEX model and assumed to be a sequel. (**b**) The same lesion with increased sonographic morphological complexity observed at the 3rd evaluation, 9 months after the diagnosis (multilocular—solid tumor with color score 3, CA-125 14.1 U/mL).

**Figure 3 diagnostics-11-01992-f003:**
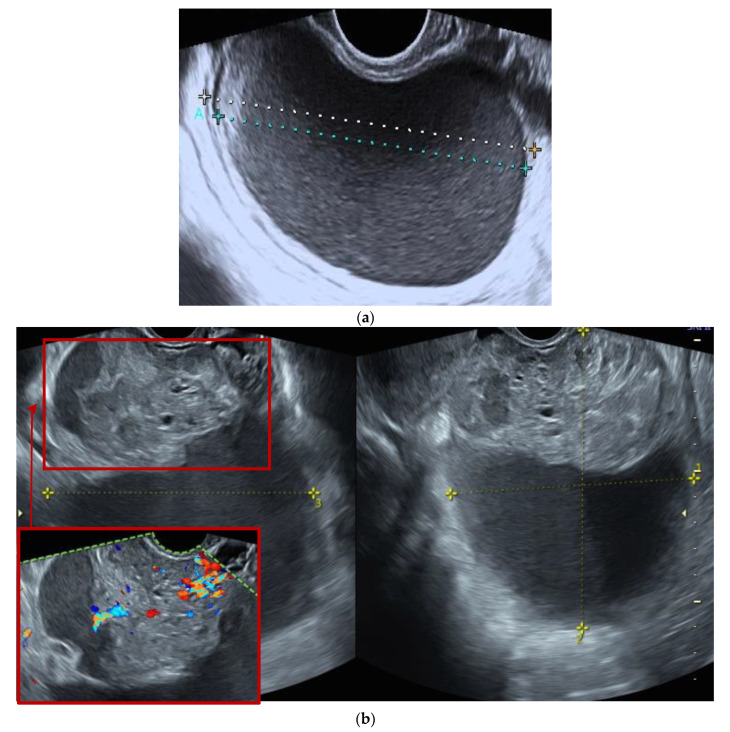
Ultrasound features of an ovarian mucinous carcinoma (FIGO IA stage) initially classified as a benign lesion. (**a**), Unilocular ovarian formation with “ground glass” content and color score 1 with 72 × 69 × 46 mm, observed in an asymptomatic pre-menopausal woman (age: 46 years), classified as benign by the IOTA ADNEX model and labeled as a possible endometrioma. (**b**) The same lesion with increased sonographic morphological complexity observed at the 2nd evaluation, 3 months after the initial diagnosis (multilocular—solid tumor with color score 3, CA-125 46 U/mL), associated with persistent pelvic pain referred by the patient as moderate with two weeks duration.

**Table 1 diagnostics-11-01992-t001:** Patient demographic and clinical characteristics.

Characteristic	Premenopausal Women, *n* (%)	Postmenopausal Women, *n* (%)
Nulliparous	126 out of 260 (48.5%)	23 out of 205 (11.2%)
Using hormonal contraception/menopausal hormonal replacement	50 (19.2)	11 (5.3)
Personal history of breast cancer	18 (6.9)	26 (12.7)
Personal history of ovarian cancer	-	-
Personal history of previous ovarian surgery	24 (9.2)	10 (4.9)
Previous Hysterectomy	14 (5.3)	29 (14.1)

**Table 2 diagnostics-11-01992-t002:** Ultrasound characteristics and classification of adnexal masses at diagnosis (1st evaluation).

Patient Group (Total Number of Masses, Percentage)	Premenopausal Women (*n* = 328, 57.2%)	Postmenopausal Women (*n* = 245, 42.8%)
**Diameter of the lesion (mm)**		
Maximum	140	135
Median (SD)	50.2 (20.7)	46.7 (19.2)
**Tumour type using IOTA terminology**		
Unilocular	202 (61.6%)	106 (43.3%)
Unilocular—solid	9 (2.7%)	12 (4.9%)
Multilocular	90 (27.4%)	90 (36.7%)
Multilocular—solid	4 (1.2%)	4 (1.6%)
Solid	23 (7%)	33 (13.5%)
**Ultrasound examiner’s subjective assessment**		
Simple, para-ovarian or salpingeal cyst	44 (13.4%)	31 (12.7%)
Serous cystadenoma	56 (23.5%)	119 (48.6%)
Mucinous cystadenoma	12 (3.7%)	5 (2%)
Endometrioma	98 (30%)	6 (2.4%)
Teratoma	39 (13.3%)	14 (5.7%)
Fibroma of fibrothecoma	21 (6.4%)	34 (13.9%)
Hydrosalpinx	40 (12.2%)	15 (6.1%)
Serous cystadenofibroma	8 (2.4%)	12 (4.9%)
Abscess, salpingitis or pelvic inflammatory disease	5 (1.5%)	5 (2%)
Inclusion or peritoneal cyst	1 (0.3%)	3 (1.2%)
Not possible to define	4 (1.2%)	1 (0.4%)

**Table 3 diagnostics-11-01992-t003:** Patient ultrasound and clinical evolution (summary of the study’s main outcomes).

Patient Group (Total Number of Masses, Percentage)	Premenopausal Women (*n* = 328, 57.2%)	Postmenopausal Women (*n* = 245, 42.8%)
Spontaneous resolution	73 (22.3%)	21 (8.6%)
Persistent mass under conservative management	187 (57%)	189 (77.1%)
Increasing complexity	12 (3.7%)	10 (4%)
Persistent adnexal mass annual growth (%), median [mean ± SD]	23.2 ± 176	10.8 ± 62
Going under surgery	68 (20.7%)	35 (14.3%)
Indication for surgery		
- Suspicion of malignant transformation	17 (25%)	11 (31.4%)
- De novo symptoms	6 (8.8%)	-
- Patient request/opportunistic	24 (35.3%)	24 (68.6%)
- Fertility concerns	21 (30.9%)	-
Complications:		
- Borderline tumour diagnosis	-	1 (0.5%) *
- Invasive malignancy diagnosis	1 (0.3%) **	-
- Adnexal mass torsion	1 (0.3%) ***	-
- Cyst rupture	-	-
No mass complications	292 (99.3%)	244 (99.6%)

* See Figure 2 for more information. ** See Figure 3 for more information. *** Adnexal torsion was observed in a 40-year-old patient with endometrioma (lesion size: 67 × 46 × 61 mm at the first and 80 × 66 × 70 mm at the second assessment, performed 3 months later).

**Table 4 diagnostics-11-01992-t004:** Spontaneous resolution of adnexal masses observed during follow-up.

Adnexal Masses with Spontaneous Resolution, (*n* = 94, 16.4%)		
Patient Group (Total Number of Masses, Percentage)	Premenopausal Women (*n* = 73, 80.2%)	Postmenopausal Women (*n* = 21, 23.1%)
**Diameter of the lesion (mm)**		
Range	31–119	34–67
Median (± SD)	45.6 ± 16.4	42.3 ± 12.3
**Time interval to resolution (months) (median ± SD)**		
Median ± SD	11 ± 10	18.6 ± 10
In the first year of follow-up	41 (56.2%)	5 (23.8%)
**Tumour type using IOTA terminology**		
Unilocular	42 (57.5%)	6 (33.3%)
Unilocular—solid	-	-
Multilocular	30 (41.1%)	15 (83.3%)
Multilocular—solid	-	-
Solid	-	-
**Ultrasound examiner’s subjective assessment**		
Simple, para-ovarian or salpingeal cyst	23 (31.5%)	3 (14.3%)
Serous cystadenoma	17 (23.3%)	6 (28.6%)
Mucinous cystadenoma	1 (1.4%)	-
Endometrioma	10 (13.7%)	2 (9.5%)
Teratoma	-	-
Hydrosalpinx or salpingitis	15 (20.5%)	9 (42.9%)
Abscess, salpingitis or pelvic inflammatory disease	4 (5.5%)	-
Inclusion or peritoneal cyst	-	1
Serous cystadenofibroma	-	-
Not possible to define	3 (4.1%)	-

**Table 5 diagnostics-11-01992-t005:** Adnexal masses showing increased sonographic morphological complexity during follow-up.

Adnexal Masses with Increased Complexity, *n* = 22	
Presumptive Histology Class (Sonographer Subjective Assessment at 1st Evaluation)	*n* (%)
Simple, para-ovarian or salpingeal cyst	3 (13.6%)
Serous cystadenoma	11 (50%)
Mucinous cystadenoma	-
Endometrioma	2 (9%)
Teratoma	-
Fibroma of fibrothecoma	-
Hydrosalpinx	3 (13.6%)
Serous cystadenofibroma	-
Abscess, salpingitis or pelvic inflammatory disease	2 (9%)
Inclusion or peritoneal cyst	-
Not possible to define	1 (4.5%)

**Table 6 diagnostics-11-01992-t006:** Histological diagnosis of surgically removed adnexal masses.

Adnexal Masses Going Under Surgery, (*n* = 103, 18%)	
Simple, paraovarian or parasalpingeal cyst	3 (2.9%)
Endometrioma	19 (18.4%)
Teratoma	14 (13.6%)
Serous cystadenoma	27 (26.2%)
Mucinous cystadenoma	8 (7.8%)
Fibroma	10 (9.7%)
Hydrosalpinx or salpingitis	8 (7.8%)
Peritoneal pseudocyst	1 (1%)
Brenner tumour	1 (1%)
Serous cystadenofibroma	9 (8.7%)
Mucinous cystadenofibroma	1 (1%)
Invasive malignancy	1 (1%)
Borderline tumour	1 (1%)

## Data Availability

The data presented in this study are available on request from the corresponding author. The data are not publicly available due to data protection.

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
