# Peer review of "Conservative Management of Asymptomatic Adnexal Masses Classified as Benign by the IOTA ADNEX Model: A Prospective Multicenter Portuguese Study"

_diagnostics, 2021, doi:10.3390/diagnostics11111992_

Round 1
Reviewer 1 Report
The authors conducted a multicentric Portuguese project aimed to observe the US morphological evolution of asymptomatic adnexal masses diagnosed as benigh according to the IOTA ADNEX model in pre and postmenopausal women, in order to assess symptoms apperance and incidence of complications during clinical and sonographic follow-up.
Although the issue could be potentially interesting, several minor criticisms have to be raised and some revisions are required in order to improve the quality of the manuscript: in particular
General point
-
A careful revision and update of literature should be considered important to improve the quality of the paper.
-
A wide revision of English language and punctuation, a proper formatting of the text and the tables should be considered mandatory to improve the quality of the paper.
-
The methods section seems to be short and not accurate.
-
The results section seems to be checked.
-
The discussion section seems to be redundant.
Besides these major concerns, other issue have to be raised and some typing mistakes exists. I have some comments.
Specific points:
Major comments:
1) INTRODUCTION
- The introduction section seems to be short and not accurate. Please try to reach out clinical features.
2) MATERIAL AND METHODS
- Use some abbreviation that are not explained in the proper section. Please modify and correct.
- Page 2 line 74 Which is the argument for choosing January 2016 as starting year?
- Page 3 How many years of experience in gynecological ultrasound have the ultrasound examiners? Are they gynecologist dedicated? Are they gynaecologist dedicated radiologist? Or general radiologist? Please clarify and add in the text.
- How many ultrasound examiners of the authors perform the ultrasound scan in the protocol?
- How were the ultrasound results obtained: exclusively by reading the ultrasound reports or retrieving information from the ultrasound databases, i.e. no reviewing of images or videoclips? When evaluating the images did also have access to the original ultrasound report? What happened if there was disagreement between what was written in the ultrasound report and what those looking at the images noted?
- The whole description of how ultrasound information was retrieved must be re-written, because from the current text it is not even clear that images and or clips had been saved electronically and that these electronic images and or were used for analysis.
- In how many cases did you used the transabdominal approach and in how many only the transvaginal? Please specify.
- Do you have any information about CA125. A comment on this would be appreciated.
- Do any of the patients also have MRI or CT or PET-CT scan and did the MRI or CT or PET_CT scan contribute at all to the correct diagnosis. Please comment on it.
- Page 4 How many years of experience in gynecological pathology have the pathologist examiners? Please clarify.
The methodology is essential and needs to be described in more detail. All the above must be made absolutely clear in the methods section.
It is quite crucial how your cases were selected. I think that you should comment on this in this section.
3) RESULTS
- Page 4 The matching between the data and the percentage in the text and the tables should be carefully revised.
- Page 7 How big was the cyst complicated by isolated torsion and what kind of cyst was? A comment on it would be appreciated.
4) DISCUSSION
The conclusions session seems to be to be redundant.
There is no need to repeat the most important results as a conclusion in the end of manuscript. The most important results should be summarized in the first paragraph. In the last paragraph I would like to see a more detailed comment on the need for further studies. Would it be meaningful to perform multicentric studies in order to get a large number of cases?
5) FIGURE LEGENDS AND FIGURES
- Please use figure with higher resolution. Please modify.
6) REFERENCES
- Considering the authors guidelines please check carefully and correct the punctuation..
- The matching between the text and the references should be carefully revised.
Author Response
Revision Note:
Dear Editor and Reviewers,
We do appreciate your suggestions and the time you have devoted to our manuscript. We have accepted all your suggestions. Please, receive our revised manuscript, revised manuscript showing the changes, as well as this Revision Note, which serves as a letter detailing the revisions.
The Authors
Reviewer general point:
- A careful revision and update of literature should be considered important to improve the quality of the paper.
Response: A careful revision and update of literature have been provided in the new version of the Introduction.
- A wide revision of English language and punctuation, a proper formatting of the text and the tables should be considered mandatory to improve the quality of the paper.
Response: Done.
- The methods section seems to be short and not accurate.
Response: Improved (please, see the responses to the specific points).
- The results section seems to be checked.
Response: Done.
- The discussion section seems to be redundant.
Response: Improved (please, see the responses to the specific points).
Reviewer specific points:
Major comments:
1) INTRODUCTION
- The introduction section seems to be short and not accurate. Please try to reach out clinical features.
Response: A careful revision and update of literature have been provided in the new version of the Introduction.
2) MATERIAL AND METHODS
- Use some abbreviation that are not explained in the proper section. Please modify and correct.
Response: Done.
- Page 2 line 74 Which is the argument for choosing January 2016 as starting year?
Response: We provided the information. “It (the study) included ≥18-year-old non-pregnant patients with asymptomatic adnexal masses (associated risk of malignancy < 10% as determined by Assessment of Different NEoplasias in the adneXa [ADNEX] model), sonographically diagnosed during an arbitrarily chosen five-year period from January, 2016 until December, 2020.”
- Page 3 How many years of experience in gynaecological ultrasound have the ultrasound examiners? Are they gynecologist dedicated? Are they gynaecologist dedicated radiologist? Or general radiologist? Please clarify and add in the text.
Response: We specified. “All ultrasound examinations were performed by five gynaecology specialists, professionally predominantly dedicated to gynaecological ultrasound scanning and certified by the International Ovarian Tumor Analysis (IOTA) group (DD, MJB, PA, PP, RC; average experience in gynaecology ultrasound: 11 years, range 5 – 30 years).”
- How many ultrasound examiners of the authors perform the ultrasound scan in the protocol?
Response: We specified. “All ultrasound examinations were performed by five gynaecology specialists, professionally predominantly dedicated to gynaecological ultrasound scanning and certified by the International Ovarian Tumor Analysis (IOTA) group (DD, MJB, PA, PP, RC; average experience in gynaecology ultrasound: 11 years, range 5 – 30 years).”
- How were the ultrasound results obtained: exclusively by reading the ultrasound reports or retrieving information from the ultrasound databases, i.e. no reviewing of images or videoclips? When evaluating the images did also have access to the original ultrasound report? What happened if there was disagreement between what was written in the ultrasound report and what those looking at the images noted?
Response: We specified. “All ultrasound data were obtained by reading electronic ultrasound reports whose accuracy was supported by accompanying images and/or videos, stored in the hospital ultrasound databases, without detected cases of discrepancies between the images/videos and the descriptions.”
- The whole description of how ultrasound information was retrieved must be re-written, because from the current text it is not even clear that images and or clips had been saved electronically and that these electronic images and or were used for analysis.
Response: We provided the information. “All ultrasound data were obtained by reading electronic ultrasound reports whose accuracy was supported by accompanying images and/or videos, stored in the hospital ultrasound databases, without detected cases of discrepancies between the images/videos and the descriptions….”
- In how many cases did you used the transabdominal approach and in how many only the transvaginal? Please specify.
Response: We provided the information. “In all cases, the scan was performed with a transvaginal and also an abdominal probe, while for the purposes of the study, only measures of adnexal masses obtained by the transvaginal approach were taken into account.”
- Do you have any information about CA125. A comment on this would be appreciated.
Response: This information already exists in the manuscript. “At each follow-up visit, presence of symptoms was assessed, physical examinations performed, transvaginal and abdominal ultrasound scan executed and blood cancer antigen 125 (CA-125) level determined.” We also included the following information: “The CA-125 levels were included in the ADNEX model for determining the risk of malignancy.”
- Do any of the patients also have MRI or CT or PET-CT scan and did the MRI or CT or PET_CT scan contribute at all to the correct diagnosis. Please, comment on it.
Response: We included the information: “According to the hospital protocol, whenever malignancy was suspected, the patient was examined by thoracic, abdominal and pelvic computed tomography (CT) before the surgery in order to plan an appropriate management.” The effectiveness of non-sonographic imaging was not evaluated, as it would exceed the study objectives.
- Page 4 How many years of experience in gynaecological pathology have the pathologist examiners? Please clarify.
Response: We included the information: “Specimen histological examination was performed in all cases by two experienced pathologists of the host hospitals (both over 15 years of specialist experience).”
The methodology is essential and needs to be described in more detail. All the above must be made absolutely clear in the methods section.
Response: We have provided all the information requested above.
It is quite crucial how your cases were selected. I think that you should comment on this in this section.
Response: We specified: "It (the study) included ≥18-year-old non-pregnant patients with asymptomatic adnexal masses (associated risk of malignancy < 10% as determined by Assessment of Different NEoplasias in the adneXa [ADNEX] model), sonographically diagnosed during an arbitrarily chosen five-year period from January, 2016 until December, 2020. Of all the asymptomatic cases consecutively observed during this period, only the patients with de novo diagnosed masses were included... Patients participating in other studies and those with evident physiological formations < 3 cm were excluded (see Figure 1)."
3) RESULTS
- Page 4 The matching between the data and the percentage in the text and the tables should be carefully revised.
Response: Corrected.
- Page 7 How big was the cyst complicated by isolated torsion and what kind of cyst was? A comment on it would be appreciated.
Response: The information was added; see Table 3. “Adnexal torsion was observed in a 40-year-old patient with endometrioma (lesion size: 67 x 46 x 61 mm at the first and 80 x 66 x 70 mm at the second assessment, performed 3 months later).”
4) DISCUSSION
The conclusions session seems to be to be redundant.
There is no need to repeat the most important results as a conclusion in the end of manuscript. The most important results should be summarized in the first paragraph. In the last paragraph I would like to see a more detailed comment on the need for further studies. Would it be meaningful to perform multicentric studies in order to get a large number of cases?
Response: Improved. Please see the new version of the Discussion.
5) FIGURE LEGENDS AND FIGURES
- Please use figure with higher resolution. Please modify.
Response: Provided.
6) REFERENCES
- Considering the authors guidelines please check carefully and correct the punctuation.
Response: Checked and corrected by using EndNote software.
- The matching between the text and the references should be carefully revised.
Response: Checked and revised.
Reviewer 2 Report
I recommend minimal shape adjustments!
I recommend minimal form corrections: a more succinct presentation of the results and discussions!Author Response
Revision Note:
Dear Editor and Reviewers,
We do appreciate your suggestions and the time you have devoted to our manuscript. We have accepted all your suggestions. Please, receive our revised manuscript, revised manuscript showing the changes, as well as this Revision Note, which serves as a letter detailing the revisions.
The Authors
“I recommend minimal form corrections: a more succinct presentation of the results and discussions!”
Response: Done, a careful revision and update of discussion was done in order to improve the quality of the paper.